# Study protocol: Dental and periodontal characteristics of older adults in Atahualpa, Ecuador within the Atahualpa Project cohort

Marcelo Armijos Briones [1]*, Patricia Estefanía Ayala Aguirre[1],
Pablo Lenin Benitez Sellan[1], Antonio Lanata-Flores[1], Oscar Marcillo-Toala[1],
Denisse A. Rumbea[2], Robertino M. Mera[3], Oscar H. Del Brutto[2]

1 School of Dentistry, Universidad de Especialidades Espíritu Santo, Samborondón, Ecuador, 2 Research Center, Universidad Espíritu Santo, Samborondón, Ecuador, 3 Biostatistics/Epidemiology, Freenome, Inc., South San Francisco, California, United States of America

* fernandoarmijos@uees.edu.ec

## Abstract

This study protocol describes an observational, cross-sectional investigation of oral health in community-dwelling older adults from Atahualpa, a rural village in Ecuador, embedded within the long-standing Atahualpa Project cohort. Using a door-to-door survey approach, all registered residents aged ≥60 years (n = 410) will be invited to participate. Participants will attend twice weekly at the School of Dentistry of University through six coordinated stations: The first station will include a general oral diagnosis and the collection of general participant data using the unique Atahualpa project identifier. The second station will record the periodontal clinical examination according to the AAP/EFP 2018 clinical guidelines. In addition, subgingival biofilm sampling will be performed for 16S rRNA sequencing and gingival crevicular fluid for multiplex cytokine profiling. The third station will obtain high-resolution digital models through an intra-oral scanning (IOS) for dental morphometrics. The fourth station will record a cone-beam computed tomography (CBCT) to support periodontal charting, and complementary structural findings. The fifth station will indicate dental treatments: restorations, root canals, extractions, and prosthesis using computer-aided design and computer-aided manufacturing (CAD/CAM). Finally, the last station will perform an oral health education: oral health promotion and structured questionnaires such as the Geriatric Oral Health Assessment Index (GOHAI), the European Health Literacy Survey-16 (HLS-EU-Q16) (in its Spanish version) and the Short Health Literacy Screen (BHLS). Primary objectives are to quantify associations between periodontitis severity, microbial composition, and local inflammatory profiles with neurocognitive and neuroimaging outcomes already available in the cohort (total MoCA and a composite cerebral small vessel disease score), controlling. Therefore, this protocol addresses the research question of whether, in older adults residing in the Atahualpa community, greater periodontitis severity is associated with lower

**Data availability statement:** Deidentified research data will be made publicly available when the study is completed and published.

**Funding:** Freenome, Inc. as the funder, providing salary support to Robertino M. Mera only, with no grant number. Freenome, Inc. did not participate in the study design, data collection, data analysis, decision to publish, or preparation of the manuscript.

**Competing interests:** The authors declared no competing interest. RMM's employment at Freenome, Inc., and to confirm that this affiliation does not alter our adherence to PLOS ONE policies on data and materials sharing. There are no patents, products in development, or marketed products related to this research that require disclosure. This study protocol is not associated with any commercial product or intellectual property.

neurocognitive performance and a higher burden of cerebral small vessel disease on neuroimaging within the existing cohort. It also examines whether the composition of the subgingival microbiome and cytokine profiles in gingival crevicular fluid correlate with periodontal severity and are related to these cerebral outcomes. Furthermore, it describes three-dimensional dental morphology (including features derived from ASUDAS for comparison with published Pima frequencies) to clarify whether there is similarity between the two ethnic groups.

## Introduction

Oral health is a fundamental dimension of overall well-being and constitutes a key indicator of population health status [1]. In rural and underserved communities, such as Atahualpa, located in coastal Ecuador, oral diseases often remain underdiagnosed and untreated, despite their high prevalence and association with systemic conditions [2,3]. The Atahualpa population is a low-income rural community, with limited access to basic services and one multidisciplinary health center staffed with two general physicians, two nurses, one dentist, and one obstetrician. The populations has sustained itself through agrarian work, craftsmanship and informal economy. Most residents have a Primary educational level (education basically), and many families live in houses made of cane or wood. The Atahualpa Project has revealed relevant data on the prevalence of non-communicable diseases within the rural community of Atahualpa, Ecuador. Through clinical examinations, the study has identified cardiovascular risk factors such as obesity, hypertension, and poor diet, highlighting the importance of implementing public health strategies to address these problems in the population [4]. The project is a community-based epidemiological initiative and has generated valuable evidence on various chronic conditions in this population. However, the oral health dimension of the inhabitants has not been comprehensively studied.

A previous study has characterized relevant aspects of the epidemiological profile of this community, including cardiovascular, neurodegenerative, and infectious diseases. Despite these advances, the morphological and pathological characterization of the oral cavity, as well as the identification of periodontopathogen microorganisms, have remained unexplored areas in this population. Within the same cohort, severe tooth loss, often regarded as an indirect marker of chronic periodontitis, has been associated with worse cognitive performance and with higher stroke incidence, suggesting potential oral–brain–vascular pathways that merit targeted investigation [5,6].

Although the biological pathways underlying these associations are not yet fully clarified, chronic periodontitis may contribute to systemic inflammation through the release of pro-inflammatory cytokines, which could negatively affect brain function [7–10]. In addition, elevated systolic blood pressure and increased pulse pressure have been associated with severe tooth loss in this population, underscoring the association between cardiovascular health and oral health [6].

In addition to the main epidemiological objective, an exploratory anthropological comparison will be conducted, there is a clear biological rationale for comparing

Atahualpa with the Pima population. Recent genomic work in a representative sample of the Atahualpa population shows a 94.1% of Indigenous American ancestry and a distinctive local genetic profile, consistent with long-standing residence and adaptation in this coastal setting [11]. Both Atahualpa residents and North American Indigenous groups such as the Pima ultimately descend from the "First Americans," who derived from Northeast Asian ancestors that dispersed into the Americas via Beringia during the Late Pleistocene; this deep shared ancestry is robustly supported by palaeogenomic and population-genetic evidence [12,13]. The Pima are among the best-characterized Indigenous populations in dental anthropology, with classic Arizona State University Dental Anthropology System based reports documenting high frequencies of specific non-metric crown traits, which makes them a natural benchmark for population-level dental comparisons [14]. While digital intraoral datasets are not yet available for the Pima, their published trait frequencies allow rigorous frequency-level contrasts today, and Atahualpa's scans will create a reproducible baseline for future, shape-based comparisons as compatible datasets emerge [15].

The primary objective of this study is to explore the association between periodontal disease severity and neurocognitive as well as neuroimaging outcomes in older adults from Atahualpa. Secondary objectives include characterizing subgingival microbiome and cytokine profiles, assessing 3D dental morphology and evaluating oral health related quality of life and health literacy. To this end, clinical diagnoses of oral diseases will be performed, including the collection of gingival samples for microbiological analysis of periodontitis-associated bacteria and inflammatory cytokines. This comprehensive characterization will help establish the actual burden of oral disease in the population, identify associated risk factors, and explore correlations with other chronic conditions previously documented in the cohort.

This protocol integrates oral health into the Atahualpa Project to build a comprehensive model of aging and disease in rural Ecuador.

## Materials and methods

### Study design and setting

This is an observational, cross-sectional study in community-dwelling residents of Atahualpa, Ecuador. This is a rural village previously characterized by the Atahualpa Project, which provides a high-quality census-based database covering more than 12 years of follow-up [4].

### Population and eligibility

According to the project's household census, 410 residents are aged ≥60 years. All will be invited to participate if they meet the following criteria: age ≥ 60 years, permanent residence in Atahualpa, active registry in the Project's database, and signed informed consent. Fully edentulous individuals will be excluded from the morphometric characterization and from sampling of periodontal pockets and gingival crevicular fluid. No probabilistic sampling is necessary because the whole population of the Atahualpa project will be targeted; this survey approach maximizes internal validity for the source population and avoids sampling variance.

### Sample size and power considerations

This is a census-based survey of the Atahualpa Project older-adult population; all 410 residents aged ≥60 years will be invited to participate. For descriptive estimates, assuming p = 0.50, a 95% confidence level and 5% absolute precision with finite population correction (N = 410), the minimum required sample is approximately 199 participants. For the primary association analyses, this sample size provides 80% power (two-sided α = 0.05) to detect small effects on continuous outcomes (e.g., correlations around r 0.20); if participation approaches the full census (n = 410), detectable effects decrease to roughly r 0.17. We will prioritize prespecified covariates and dimension reduction for biomarker panels to maintain model stability.

It is important to note that data is not currently being collected for this protocol; it is expected to begin in March 2026 and end in November of the same year.

## Clinical workflow and data capture

Data collection will take place at the School of Dentistry Clinic of our Institution (48 dental chairs), enabling simultaneous care across six stations: (1) oral diagnosis diagnostics, (2) periodontics, (3) intraoral 3D scanning for dental morphometrics, (4) 3D imaging (cone beam computed tomography), (5) indicated dental treatments, and (6) classroom-based counseling plus interviews/questionnaires. Upon entry (station 1), each participant's unique Atahualpa Project ID will be verified; newly collected clinical variables are appended to the existing raw data to preserve longitudinal traceability.

**Station 1: General oral diagnosis.** An intake form will record relevant dental history, oral hygiene habits, tobacco use, and medications; a basic intraoral exam will be performed (soft tissues, present/missing teeth, mobility, visible furcation exposures). The intake form will specifically capture recent exposures that may affect the oral microbiome, including systemic antibiotics and antifungals, topical oral antimicrobials (e.g., chlorhexidine), and other relevant agents (e.g., immunosuppressants). We will record timing, duration, and indication for these medications.

Plaque index and other hygiene indicators will be recorded in a standardized manner for subsequent analysis. Dental plaque will be assessed using the O'Leary Plaque Control Record (PCR). After applying a disclosing solution, visible plaque will be recorded as present/absent on four surfaces per tooth (mesial, distal, facial/buccal, and lingual/palatal) for all teeth present. The PCR will be expressed as the percentage of tooth surfaces with plaque (number of surfaces with disclosed plaque divided by the total number of surfaces examined ×100). Missing teeth will be excluded from the denominator. Oral hygiene (PCR) will be incorporated as a prespecified covariate in analyses of periodontal inflammation and related biomarker outcomes. This newly acquired data will complement existing cohort information and align with the Project's in-person measurement standards.

All oral-diagnosis assessments will be performed by trained dentists following a written standard operating procedure. Prior to fieldwork, examiners will complete a calibration workshop with a pilot series of participants to standardize examination sequence and recording criteria; a subset of participants will undergo repeat examination to estimate intra- and inter-examiner agreement for key recorded items (e.g., tooth presence/absence, mobility and visible furcation exposure), with retraining if agreement is below prespecified targets.

**Station 2: Periodontal clinical examination and biological sampling.** Periodontal assessment will follow a sequence of conditioning, full-mouth clinical charting, and diagnosis under the American Academy of Periodontology (AAP) land the European Federation of Periodontology (EFP) 2018 framework, later integrating the CT scan findings from Station 4 [16].

*Conditioning and clinical exam*

Before measurements, a limited supragingival debridement will be performed to standardize visualization, with relative isolation and gentle drying. The periodontal charting includes six sites per tooth probing depth, bleeding on probing, Clinical Attachment Loss (CAL), furcation involvement, and suspected vertical defects. Bone loss will be confirmed with CT scans. Periodontitis will be defined as interdental clinical attachment loss (ICL) in ≥2 non-adjacent teeth, or buccal or lingual ICL ≥ 3 mm with a probing depth >3 mm in ≥2 teeth, after excluding non-periodontal causes [16,17]. The stages and grades will be measured as described in Table 1.

Full-mouth periodontal charting will be performed on all teeth present (excluding third molars, if applicable) at six sites per tooth (mesiobuccal, mid-buccal, distobuccal, mesiolingual/palatal, mid-lingual/palatal, distolingual/palatal) using a manual periodontal probe. Probing depth (PD, mm) will be measured from the gingival margin to the base of the sulcus/pocket. Gingival recession (REC, mm) will be measured from the cemento-enamel junction (CEJ) to the gingival margin, and clinical attachment level (CAL, mm) will be calculated as CAL = PD + REC (with negative values handled according to standard conventions when the gingival margin is coronal to the CEJ). Bleeding on probing (BOP) will be recorded

**Table 1. Standards for determining the stage and degree of periodontal disease.**

| Parameter | Stage | | | |
|---|---|---|---|---|
| | I | II | III | IV |
| Interdental clinical attachment loss | 1 - 2 mm | 3 - 4 mm | > 5 mm | |
| Bone loss measured on CT scan (%) | <15% | 15 - 30% | Extension to the middle or apical third of the root | |
| Tooth loss | There is no tooth loss due to periodontitis. | | Tooth loss due to periodontitis ≤ 4 | Tooth loss due to periodontitis ≥ 5 |
| Parameter | Grading | | | |
| Indirect evidence of progression | A | B | C | |
| bone-loss/age | <0.25 | 0.25–1.0 | >1.0 | |

as present/absent at each site within a standardized observation window after probing. Furcation involvement will be recorded for multirooted teeth using an established ordinal classification, and tooth mobility will be recorded as described in Station 1. Periodontitis staging/grading will follow the AAP/EFP framework; tomographic bone loss from CBCT will be used to support and refine staging by estimating the percentage of radiographic bone loss relative to root length on standardized views.

Tomographic (CBCT-derived) bone loss percentage will be operationalized as the distance from the cemento-enamel junction (CEJ) to the alveolar crest divided by the distance from the CEJ to the root apex, expressed as a percentage, assessed on standardized CBCT planes.

*Evaluation of the Composition of the Oral Microbiota*

After periodontal charting, biological sampling will be performed for microbiology and cytokines. For this, two permanent incisors and two molars will be selected; at each tooth, the deepest site will be sampled. Supragingival biofilm will be removed (the field isolated with cotton rolls); in the absence of any of these target teeth, the nearest available tooth will be selected following the same predefined criteria. We will target the deepest site per selected tooth because deeper pockets concentrate the subgingival biofilm most relevant to periodontitis pathobiology and maximize the likelihood of capturing dysbiotic periodontal communities. This approach also standardizes sampling across participants and improves feasibility in a field-based, station workflow. To obtain a participant-level profile rather than a single-site snapshot, samples from four teeth (two incisors and two molars) will be pooled, thereby integrating information across different tooth types and quadrants. Probing depth and bleeding on probing at sampled sites will be recorded to contextualize microbial profiles and support sensitivity analyses. In participants without periodontal pockets meeting the sampling criteria, the deepest available sulcus sites will be sampled using the same standardized procedure.

A sterile #35 paper point (WaveOne Gold, Maillefer Dentsply Sirona, Okla, U.S.A.) will be inserted to the base of the pocket or sulcus for 30 s. The four points will be pooled in a sterile Eppendorf tube containing 300 μL of 0.5 mM Tris–EDTA and stored at −80°C (Thermo Fisher Scientific TD600, NC, USA). DNA extraction is performed using the Qiagen MiniAmp Kit (Qiagen, Valencia, CA, U.S.A.), following the manufacturer's instructions, including an initial step to solubilize biofilm (phosphate-buffered saline + gentle agitation). 16S rRNA sequencing libraries will be prepared according to Illumina's 16S Metagenomic Sequencing Library Preparation guide and sequenced on an Illumina MiSeq, targeting ~6 million reads per participant [18]. To minimize bias in microbiome profiling, recent use of systemic antibiotics or antifungals and topical oral antimicrobials will be considered in the analytic plan. We will (i) document these exposures systematically and (ii) perform sensitivity analyses excluding participants with recent antimicrobial/antifungal exposure (e.g., within the prior 3 months) and/or adjust models for these exposures as prespecified covariates.

*Evaluation of Biomarkers in Gingival Crevicular Fluid*

 

From the same sites, after ≥30 s from biofilm collection, Gingival crevicular fluid (GCF) will be obtained using absorbent strips (Periopaper, Oraflow, Plainview, NY, U.S.A.) placed for 15 s in the sulcus or pocket. Strips will be stored in 100 µL PBS with 0.05% Tween-20 at −80 °C (Thermo Fisher Scientific TD600, NC, USA).

Cytokine profiling will be performed using a multiplex bead-based immunoassay (Luminex xMAP). The following cytokines will be quantified: IFN-γ, IL-17, IL-10, IL-1β, IL-8, IL-4, IL-6, and TNF-α, using the Sensitivity Human Cytokine 08-plex Kit (Millipore Corporation, Billerica, MA, U.S.A.). Assays will be run on the MAGpix™ system (MiraiBio, Alameda, CA, U.S.A.), following the manufacturer's instructions [19,20].

Periodontal examiners will be calibrated prior to data collection through a structured training and pilot exercise to standardize probing technique and diagnostic criteria. Inter- and intra-examiner reliability will be evaluated in a subset of participants using duplicate full-mouth recordings (e.g., within 7–14 days) for continuous measures (probing depth and clinical attachment loss; assessed with intraclass correlation coefficients) and categorical variables (bleeding on probing, furcation involvement and stage/grade assignments; assessed with kappa statistics). If agreement falls below prespecified thresholds, additional retraining and repeat calibration will be performed before continuing data collection.

**Station 3: Intra-oral scanning for dental morphometrics.** High-resolution digital impressions will be acquired with an intraoral scanner (TRIOS Core 3Shape, Copenhagen, Denmark) after a pre-field operator calibration (positioning, scan path, export parameters) and a pilot series to check inter-operator reliability. Full maxillary scans will be exported as Standard Triangle Language (STL). For geometric analysis, the right maxillary first molar will be digitally segmented into 3D Slicer (v5.0); meshes are preprocessed in MeshLab to smooth artifacts that could bias landmark placement. Anatomical landmarks (8–15 per molar; cusp tips, fossae, central groove path, cervical margin) will be placed by two trained examiners to estimate interobserver agreement. Landmark coordinates will undergo Generalized Procrustes Analysis to remove variation due to size, orientation, and position, followed by Principal Components Analysis (PCA) to summarize dominant axes of shape variation. Where relevant, group contrasts will be performed using MANOVA and Mahalanobis distances. Analyses will be conducted in MorphoJ and in the R package *geomorph*, following established geometric morphometrics practice [21].

**Morphometric comparison with the Pima population.** Classic anthropology on the Pima reported high frequencies of several non-metric crown traits (e.g., incisor shoveling, hypocone, protostylid, cusp 6), with intermediate Carabelli/cusp 7 frequencies, using the ASUDAS system [14,22]. In Atahualpa, recent genomics suggests predominantly Indigenous (Amerindian) ancestry and distinctive local features biological context for benchmarking against a well characterized Indigenous reference. We will translate Atahualpa's scans into ASUDAS states through quantitative rules (curvature, relative areas/volumes, cusp heights, groove connectivity) and double-blind scoring in a subsample to ensure reproducibility. Trait frequencies will be contrasted under a beta binomial Bayesian framework; Multitrait affinity will be evaluated using Smith's Mean Measure of Divergence (MMD; implemented in the AnthropMMD R package) and PERMANOVA on a trait dissimilarity matrix (e.g., Gower), with adjustment for age, sex, and tooth wear [15,23–25].

**Station 4: Cone-beam computed tomography.** Cone beam computed tomography (CBCT) will be obtained with a Planmeca Viso G7 scanner with a bimaxillary field of view, using a low-dose protocol compatible with the assessment of alveolar ridge height/thickness, furcations, vertical defects and root trajectories. The use of CBCT is justified by its contribution to the stratification of periodontal disease and the resolution of diagnostic uncertainties not addressed with periapical radiographs, as well as by its excellent image quality, which facilitates future interethnic comparisons. All scans will have standardized positioning and artifact movement control using calibrated systems. [17].

**Station 5: Dental treatments.** Treatment will be delivered according to individualized care plans after clinical assessment: direct restoration, endodontic therapy, extractions, and prosthetic rehabilitation. Standard protocols will be applied for infection control, local anesthesia, and record-keeping linked to the cohort identifier. Adhesive restorations will use contemporary techniques with relative or absolute isolation as appropriate; endodontic therapy will follow conservative access, working length by combined clinical and radiographic methods, mechanized instrumentation, and thermoplastic

or lateral condensation obturation, with post-operative periapicals for quality checks. Extractions will be performed atraumatically with osteotomy/odontosection when required. Prosthetic rehabilitation will leverage the intraoral STL files from Station 3 for CAD/CAM workflows (removable prostheses), with documentation of functional/occlusal adjustments and patient reported fit/satisfaction where applicable.

**Station 6: Oral health education: Oral health promotion and structured questionnaires.** After clinical care, participants will be directed to a classroom on the same floor for interviews and questionnaires, followed by a group educational session on prevention and healthy habits. Selected instruments include the Geriatric Oral Health Assessment Index (GOHAI) [26], administered in updated Spanish versions [27,28]. Two GOHAI scores will be computed the Additive GOHAI (ADD-GOHAI; range 12–60; higher scores indicate better oral health–related quality of life) and the Simple-count GOHAI (SC-GOHAI; range 0–12; higher scores indicate worse outcomes), with items 3 and 7 reverse-coded. Health literacy will be measured using the European Health Literacy Survey Questionnaire, short form with 16 items (HLS-EU-Q16), and the Brief Health Literacy Screen (BHLS, 3 items), both analyzed as continuous scores in the main analyses given heterogeneous cutoffs. These instruments have demonstrated adequate psychometric performance in Spanish and English-speaking populations [26–29]. Educational content covers oral hygiene, plaque control, home fluoridation, and removable-prosthesis care (daily non-abrasive cleaning, nighttime removal, dry storage, avoidance of continuous wear, periodic checks). All these instruments will be placed in a Google Forms and then administered to participants with the help of graduate students. Data will be automatically entered into an online Microsoft Excel spreadsheet.

Clinical data will be recorded on pre-coded forms and entered into structured electronic databases using standardized variable definitions. Questionnaire data will be collected using electronic forms with built-in range and completeness checks. Data will be stored on secure institutional servers with role-based access; the analytic dataset will be de-identified and linked to the Atahualpa Project identifier, while any linkage file will be stored separately under restricted access. Routine data-quality procedures will include daily consistency checks, periodic audits, and random re-examination of a subset of participants to verify key variables.

Fig 1 shows the workflow by stations for collecting information from participants.

## Statistical analysis plan

**Primary objective and outcomes.** The main analytical objective will be to quantify associations between periodontitis severity and local biological profiles (subgingival microbiome composition and GCF cytokines) with neurocognitive and neuroimaging outcomes already available in the cohort. Primary outcomes will be cognitive impairment assessed by the MoCA and defined by the 19–20 cutoff previously proposed for this rural cohort; the neurovascular outcome is the total cerebral small vessel disease (SVD) score as a composite construct, complemented by individual MRI markers (white-matter hyperintensities, lacunes, deep microbleeds, and enlarged basal-ganglia perivascular spaces) as used in prior Atahualpa studies [4,30].

Although the protocol includes exploratory components, we specify a priori directional hypotheses for the primary objective to minimize analytic flexibility. We hypothesize that greater periodontitis severity (higher stage/grade and worse continuous periodontal measures) will be associated with poorer cognitive performance (lower MoCA) and a higher neuroimaging burden of cerebral small vessel disease (higher total SVD score). We further hypothesize that greater periodontitis severity will track with a more dysbiotic subgingival microbiome profile and a more pro-inflammatory gingival crevicular-fluid cytokine signature, and that these local biological profiles will correlate with MoCA and SVD outcomes. All remaining analyses (e.g., mediation, moderation, and morphometric comparisons) will be treated as exploratory and hypothesis-generating.

**Exposures and biomarkers.** Periodontal exposure will be represented by AAP/EFP stage and grade, plus participant-level continuous aggregates (mean/max probing depth, CAL, % bleeding sites). The percentage of tomographic bone loss on CBCT will support staging. Microbiome profiles derive α-diversity (Shannon, Chao1) and

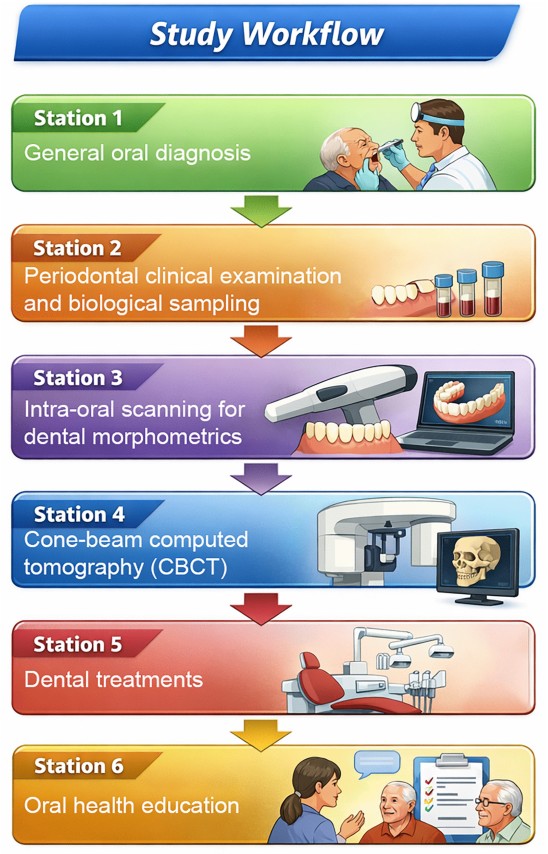

**Fig 1. Study workflow and data collection across clinical stations in the Atahualpa Project.** Participants attend six coordinated stations: (1) general oral diagnostics; (2) clinical examination and periodontal sampling; (3) intraoral 3D scanning and dental morphometrics; (4) cone beam computed tomography (CBCT); (5) indicated dental treatments; and (6) oral-health counseling, questionnaires, and interviews. Image created with AI. ChatGPT version 5.

β-dissimilarity (e.g., Bray–Curtis), with taxonomic assignment to the most resolvable level and relative abundances of periodontopathogens. GCF cytokines will be log-transformed and standardized; a local inflammatory index will be derived via PCA. Oral-health quality of life (GOHAI) and health literacy (HLS-EU-Q16; BHLS) will serve as secondary outcomes and exploratory covariates, respectively, per instrument rules.

**Morphometric comparisons.** First, trait-level frequencies will be contrasted between Atahualpa and Pima using a beta binomial Bayesian model with weakly informative Beta priors, reporting credible intervals for differences and probabilities of superiority. Second, Mean Measure of Divergence (MMD) will be computed from trait frequencies as a phenetic biodistance, with point estimates, standard errors, and significance tests (R package AnthropMMD). Third, PERMANOVA on an appropriate dissimilarity (e.g., Gower) with 9,999 permutations, adding covariates when warranted, will test global profile differences; homogeneity of dispersion will be checked beforehand [23–25].

**Main models and sensitivity analyses.** Primary analyses will model total MoCA as a continuous outcome using multivariable linear regression (assumption checks and robust options as needed) and the total cerebral small vessel disease (SVD) score (0–4) using proportional-odds ordinal logistic regression. Community-level microbiome composition will be tested using the PERMANOVA statistical test on Bray-Curtis dissimilarities, including key covariates. Periodontitis severity will enter as an ordinal exposure (Stage I–IV). In sensitivity analyses, it will be

replaced by the continuous percentage of radiographic bone loss. To limit multiplicity, gingival-crevicular cytokines will be log-transformed, standardized, and summarized into 1–2 principal component scores that enter the models instead of individual cytokines. Secondary analyses will include logistic regression for dichotomous cognitive impairment using the locally validated 19–20 MoCA cut-point, negative binomial modeling of the total SVD score if proportional-odds assumptions fail or if dispersion indicates a count framework and differential abundance testing for selected taxa using ANCOM-BC with false-discovery-rate control. Periodontal site measures will be aggregated to the participant level for all person-level outcomes. Multilevel models with sites nested within individuals will be specified only if any outcome is analyzed at the tooth or site level.

**Mediation, moderation and multiplicity.** Exploratory mediation will assess whether the effect of periodontitis severity on MoCA and SVD burden is carried by the local inflammatory index or by relative abundances of key periodontopathogens; moderation by diabetes and smoking will be examined given their role in A–C grading. Multiple testing will be controlled in related families (cytokine panels; differential taxa) using Benjamini–Hochberg FDR.

**Covariates and confounding control.** Models will be adjusted for age, sex, educational attainment, relevant social determinants of health determined by the Gijón's social-familial education scale, and traditional cardiovascular risk factors assessed by means of the Life's Essential 8 construct of the American Heart Association [30]. Oily-fish intake will be considered as a covariate relevant to Atahualpa. Tooth loss will be modeled as a key covariate or stratification variable due to its prior association with cognition and incident stroke in this population. Dietary variables relevant to Atahualpa (including oily-fish intake) are already available in the parent Atahualpa Project database and will be used as covariates in the prespecified models; this oral-health protocol does not introduce a new comprehensive dietary questionnaire.

## Ethics statement

This research project was approved by the Human Research Ethics Committee of Kennedy Clinic under code HCK-CEISH 2024−003. This review board is duly authorized to issue approvals by the Ministry of Public Health of Ecuador.

The proposed study has not yet begun data collection. Once data collection begins, it is expected that informed consent approved by the ethics committee that approved this project will be used. The consent form will be a physical document, printed on paper and handwritten and signed by each participant who agrees to participate in this project. All ethical principles will be respected to ensure the safety and well-being of the participants. There will be no cost to participants for any part of the study, nor will they receive any compensation for their participation.

Clinical and laboratory data will be captured using pre-coded forms with double checks, daily audits, and random re-examination for reproducibility. Data integrate into the Atahualpa repository using the unique identifier, preserving longitudinal continuity in line with prior field practices. The analytic dataset is anonymized; the linkage file is stored separately under secure procedures.

## Justification

Oral conditions are common yet underdiagnosed in underserved rural settings. Atahualpa, a culturally homogeneous village on Ecuador's coast offers a unique context where shared lifestyles and diet reduce unmeasured confounding and enable high-quality, door-to-door epidemiology [4,30].

Despite a decade of rigorous work in neurovascular and cognitive domains, the oral-health dimension of this cohort remains insufficiently characterized. Within the same population, severe tooth loss, often an end-stage marker of chronic periodontitis, has been associated with worse cognitive performance and with higher stroke incidence; neuroimaging studies have also documented a substantial burden of silent markers of cerebral small vessel disease (SVD), and composite SVD scores relate to cognitive performance in these older adults [5,6,30]. Taken together, these findings create a compelling rationale to quantify periodontal health, subgingival microbiology, and crevicular cytokines and to relate them to neurocognitive and neuroimaging phenotypes already available in the cohort.

The present protocol leverages a standardized, station-based clinical workflow to obtain periodontal diagnoses, bio-specimens, and high-resolution intraoral scans in concentrated visits, maximizing participant yield and internal validity. Embedding the oral assessment within the existing cohort architecture facilitates linkage with prior cardiovascular and brain-health measures and positions the study to inform prevention and care strategies tailored to rural Latin American settings [4,30].

A novel component is a comparative dental-anthropology analysis with the Pima population. Atahualpa residents show predominantly Indigenous (Amerindian) ancestry, providing biological plausibility for benchmarking non-metric crown traits against a well-documented Indigenous reference [11]. Because compatible 3D datasets are not yet available for Pima, we will derive objective ASUDAS states from Atahualpa's intraoral scans and perform frequency-level contrasts using a beta binomial Bayesian framework; to assess multitrait affinity, we will compute Smith's Mean Measure of Divergence and perform PERMANOVA on trait dissimilarities, controlling for age, sex, and tooth wear [14,22,23,25]. This approach preserves backward compatibility with the Pima literature while building a reproducible 3D baseline for future, shape-based comparisons as datasets emerge.

Beyond scientific value, the study has practical relevance. By quantifying the burden and correlating oral disease in a hard-to-reach population, the protocol could guide equitable allocation of dental and periodontal services, inform targeted health-promotion curricula, and strengthen integrated, community-based models of care that will align with the cohort's proven field methods [4,30].

## Limitations

We identified several limitations in the protocol we intend to follow. The cross-sectional design prevents establishing causal and temporal relationships. The associations between periodontitis, inflammatory profiles, microbiota, cognition, and neuroimaging should be interpreted as hypothesis-generating. The entire process described will take place in the city of Samborondón, located 128 kilometers from Atahualpa. The distance could represent a problem for participants with reduced mobility. Participants will be transported by buses funded by the project to avoid underrepresentation.

Generalization of the results is limited, as Atahualpa is a relatively homogeneous rural community with low migration rates. The findings may not be applicable to urban or ethnically diverse settings. Excluding edentulous participants for periodontal sampling could reduce the representativeness of older or more frail groups, in whom the consequences of periodontal disease may be greater.

Regarding morphometric analyses, the conversion of 3D images to ASUDAS trait states, despite predefined quantitative rules and double evaluation, introduces measurement errors and potential misclassifications. Inter- and intra-observer reliability will be reported. Comparisons with Pima are based on published trait frequencies and not on 3D data, which limits inferences at the shape level and restricts them to frequency-based statistics (e.g., beta-binomial, MMD, PERMANOVA). This allows for historical comparability but does not capture the full 3D morphology until external datasets are available. Furthermore, due to the participants' age, dental tissue wear is expected, which may interfere with morphometric measurements. Future phases of the project involving younger individuals will help improve the accuracy of comparisons.

Microbiome analysis using 16S rRNA has limited taxonomic resolution for some genera or species, is compositional, and sensitive to batch effects and variations in storage and/or processing. We will implement contamination controls, technical controls, adjust the sequencing depth, and use compositional methods (e.g., ANCOM-BC). Cytokine measurements in crevicular fluid are susceptible to pre-analytical variability (site selection, fluid volume) and cross-reactivity in multiplex assays. Strict standard operating procedures, repeated measurements, and operator calibration will be used.

The timing of the results is not perfectly synchronous. That is, the cognitive scores and MRI markers come from previous assessments of Atahualpa, which introduces temporal mismatches that may attenuate or exaggerate the associations. We will record the time intervals and perform sensitivity analyses by restricting them short periods of time.

Finally, questionnaires (GOHAI, HLS-EU-Q16, BHLS) may be subject to recall bias and social desirability bias. Trained interviewers and standardized procedures will aim to minimize these biases.

## Dissemination plans

No data were generated or analyzed for this study protocol. Upon publication of the first results derived from this protocol, the minimal de-identified dataset underlying those findings (including the variables used in the analyses) will be provided as S1 File. Direct identifiers and any linkage file will not be shared. If any specific variable is judged to pose a non-negligible risk of re-identification in this small community, that element will be withheld from public release and access to a further de-identified version will be provided upon reasonable request, subject to approval by the Human Research Ethics Committee of Kennedy Clinic (CEISH).

## Supporting information

**S1 File. English_Questionnaires_GOHAI_HLS_BHLS.**
(DOCX)

## Acknowledgments

We would like to express our gratitude to the Universidad Espíritu Santo for its support of this project, to the community of Atahualpa for its active participation in developing this protocol, and to the Research Center of the Universidad Espíritu Santo for the facilities provided.

## Author contributions

**Conceptualization:** Marcelo Armijos Briones, Patricia Estefanía Ayala Aguirre, Antonio Lanata-Flores, Denisse A. Rumbea, Robertino M. Mera-Giler, Oscar H. Del Brutto.

**Funding acquisition:** Denisse A. Rumbea.

**Methodology:** Marcelo Armijos Briones, Pablo Lenin Benitez Sellan, Oscar Marcillo Toala, Antonio Lanata-Flores, Oscar H. Del Brutto.

**Project administration:** Denisse A. Rumbea, Robertino M. Mera-Giler, Oscar H. Del Brutto.

**Writing – original draft:** Marcelo Armijos Briones, Pablo Lenin Benitez Sellan, Oscar Marcillo Toala, Antonio Lanata-Flores, Oscar H. Del Brutto.

**Writing – review & editing:** Marcelo Armijos Briones, Patricia Estefanía Ayala Aguirre, Pablo Lenin Benitez Sellan, Oscar Marcillo Toala, Antonio Lanata-Flores, Denisse A. Rumbea, Robertino M. Mera-Giler, Oscar H. Del Brutto.

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
