## [Decision Letter · Decision Letter 0]

17 Dec 2025

Dear Dr. Armijos Briones,

Thank you for submitting your manuscript to PLOS ONE. After careful consideration, we feel that it has merit but does not fully meet PLOS ONE’s publication criteria as it currently stands. Therefore, we invite you to submit a revised version of the manuscript that addresses the points raised during the review process.

Please address the comments of the reviewers and re-submit the revised manuscript.

We look forward to receiving your revised manuscript.

Kind regards,

Vinayak M. Joshi, MS, PhD

Academic Editor

PLOS One

Journal Requirements:

“The authors have declared that no competing interests exist.”

We note that one or more of the authors are employed by a commercial company: Freenome, Inc

2) Please also provide an updated Competing Interests Statement declaring this commercial affiliation along with any other relevant declarations relating to employment, consultancy, patents, products in development, or marketed products, etc.

Within your Competing Interests Statement, please confirm that this commercial affiliation does not alter your adherence to all PLOS ONE policies on sharing data and materials by including the following statement: ""This does not alter our adherence to  PLOS ONE policies on sharing data and materials.” (as detailed online in our guide for authors http://journals.plos.org/plosone/s/competing-interests) . If this adherence statement is not accurate and  there are restrictions on sharing of data and/or materials, please state these. Please note that we cannot proceed with consideration of your article until this information has been declared.

3. We note that Figure 1 in your submission contain copyrighted images. All PLOS content is published under the Creative Commons Attribution License (CC BY 4.0), which means that the manuscript, images, and Supporting Information files will be freely available online, and any third party is permitted to access, download, copy, distribute, and use these materials in any way, even commercially, with proper attribution. For more information, see our copyright guidelines: http://journals.plos.org/plosone/s/licenses-and-copyright.

1) You may seek permission from the original copyright holder of Figure 1 to publish the content specifically under the CC BY 4.0 license.

2) If you are unable to obtain permission from the original copyright holder to publish these figures under the CC BY 4.0 license or if the copyright holder’s requirements are incompatible with the CC BY 4.0 license, please either i) remove the figure or ii) supply a replacement figure that complies with the CC BY 4.0 license. Please check copyright information on all replacement figures and update the figure caption with source information.

If applicable, please specify in the figure caption text when a figure is similar but not identical to the original image and is therefore for illustrative purposes only.

**Additional Editor Comments:**

Dear authors,

Please address the reviewer comments and resubmit the manuscript with the revisions made.

Thank you.

Reviewers' comments:

Reviewer's Responses to Questions

**Comments to the Author**

1. Does the manuscript provide a valid rationale for the proposed study, with clearly identified and justified research questions?

Reviewer #1: Partly

Reviewer #2: Yes

Reviewer #3: Yes

2. Is the protocol technically sound and planned in a manner that will lead to a meaningful outcome and allow testing the stated hypotheses?

Reviewer #1: Yes

Reviewer #2: Yes

Reviewer #3: Partly

3. Is the methodology feasible and described in sufficient detail to allow the work to be replicable?

Reviewer #1: Yes

Reviewer #2: Yes

Reviewer #3: No

4. Have the authors described where all data underlying the findings will be made available when the study is complete?

Reviewer #1: No

Reviewer #2: Yes

Reviewer #3: No

5. Is the manuscript presented in an intelligible fashion and written in standard English?

Reviewer #1: Yes

Reviewer #2: Yes

Reviewer #3: Yes

You may also provide optional suggestions and comments to authors that they might find helpful in planning their study.

Reviewer #1: The topic is relevant and fits well within the context of the Atahualpa Project, and the proposed work has the potential to generate valuable data on oral and systemic health in an older population.

The rationale is appropriate, and the two objectives are clearly stated and reasonable. However, explicit research questions are not reported. I suggest that the authors add clear research questions derived from these objectives to improve clarity and alignment with the journal’s criteria.

The protocol is technically sound, and the methods are described in sufficient detail to understand the planned procedures and anticipate meaningful outcomes. The study is repeatedly described as “exploratory,” and no explicit a priori hypotheses are formulated. To improve transparency and align with the journal’s criterion regarding the ability to test stated hypotheses and avoid undisclosed analytic flexibility, I recommend that the authors either:

• clearly present formal hypotheses for their primary objective (e.g., expected direction of the association between periodontal severity and neurocognitive/neuroimaging outcomes), and/or

• explicitly state that all planned analyses are exploratory and clearly distinguish any confirmatory components from exploratory ones, if applicable.

The methodology appears feasible and is described with sufficient detail to allow replication. The setting, inclusion/exclusion criteria, examination procedures, and planned assessments are clearly outlined, and the proposed data collection procedures seem realistic within the study framework. However, some important aspects would benefit from further clarification to fully ensure reproducibility and robustness. In particular, I recommend that the authors:

• provide a clear sample size calculation or justification for the primary objective, including key assumptions; and

• describe in more detail the examiner calibration and reliability procedures for periodontal and dental assessments.

The current data availability statement is not consistent with PLOS ONE’s data policy. Stating that “results will be published in scientific journals” and that “depending on the type of data, the datasets may or may not be made publicly available” does not specify where the underlying data will be deposited nor under what conditions they will be accessible. I recommend that the authors revise this section to clearly describe any justified restrictions (e.g., privacy/identifiability in a small community) and how qualified researchers can request access to de-identified data if full open release is not possible.

The manuscript is generally written in clear and intelligible English, and the content is easy to follow. However, there are several typographical and minor grammatical errors throughout the text. I recommend a careful proofread (or professional language editing, if available) to correct these issues and ensure the manuscript meets the journal’s standard for clarity and correctness.

Reviewer #2: A well-constructed and feasible study protocol. It will add to the available data regarding Atahualpa.

Reviewer #3: I read this study protocol with great interest. It is a great study to be performed. However, it is poor described and needs improvement.

1 - How do the authors will ensure power for the statistical analysis?

2 - Since the authors want to see microbiome effects, they need to screen the use of antimicrobial agents, and anti fungal.

3 - Since it is a cohort study assessment of quality of life, as well as dietary questionnaire have to be performed.

4 - How does plaque index will be performed? This should be better described.

5 - For all clinical measurements, please, describe in more depth.

6 - Why microbial community will be assessed only deep site?

7 - Data managing and sharing. should be clearer.

**Do you want your identity to be public for this peer review?** For information about this choice, including consent withdrawal, please see our Privacy Policy

Reviewer #1: No

Reviewer #2: No

Reviewer #3: No

---

## [Author Response · Author response to Decision Letter 1]

24 Jan 2026

Editor's comments

1. Ensure the manuscript complies with PLOS ONE style requirements and file-naming conventions upon resubmission.

Thank you for the reminder. We have revised the manuscript to conform to PLOS ONE style requirements and will follow PLOS ONE file naming conventions upon resubmission (clean manuscript, tracked-changes version, and response to reviewers)

2. Revise the Funding Statement to disclose the commercial affiliation and include the required PLOS wording (salary support + no additional role + roles described in Author Contributions).

We have updated the Funding Statement to declare commercial affiliation. Specifically, we state that all authors are salaried employees of Universidad de Especialidades Espíritu Santo, except RMM, who is an employee of Freenome, Inc. We also include the requested wording indicating that the funder provided support in the form of salary for RMM only, and that the funder had no additional role in the study design, data collection and analysis, decision to publish, or preparation of the manuscript. The specific role of RMM is described in the Author Contributions section

3. Revise the Competing Interests Statement to disclose the commercial affiliation and include the required sentence: “This does not alter our adherence to PLOS ONE policies on sharing data and materials.”

We have updated the Competing Interests Statement to declare that RMM is an employee of Freenome, Inc. We also confirm, using the wording requested by PLOS ONE, that this does not alter our adherence to PLOS ONE policies on sharing data and materials.

4. Figure 1 contains copyrighted content; provide written permission compatible with CC BY 4.0 or remove/replace the figure with one that is CC BY compliant.

Thank you for highlighting this issue. We have removed the original Figure 1 and replaced it with a newly created fully original schematic workflow figure. The revised figure was generated specifically for this manuscript and does not contain any copyrighted material, third-party images, photographs, logos, or proprietary content. The new figure is fully compliant with the Creative Commons Attribution (CC BY 4.0) license required by PLOS ONE and may be freely reused, distributed, and adapted with attribution. The figure caption has been updated accordingly.

5. If reviewers suggested citations, evaluate and cite only if relevant (not automatically required).

“We carefully reviewed the reviewers’ comments; no specific additional citations were requested/required.”

6. Address all reviewer comments and resubmit a revised manuscript (tracked-changes + clean version) and a point-by-point response.

We thank the Academic Editor and reviewers for their constructive feedback. We have addressed all reviewer comments point-by-point in the response letter and revised the manuscript accordingly. A tracked-changes version and a clean version of the revised manuscript will be provided upon resubmission.

Reviewer 1

1. “…However, explicit research questions are not reported. I suggest that the authors add clear research questions derived from these objectives to improve clarity and alignment with the journal’s criteria.”

Thank you for this suggestion. We agree that explicitly stating the research questions improves clarity and alignment with the study objectives. Accordingly, we added a concise paragraph at the end of the Introduction that articulates the main research questions derived from the primary and secondary objectives, including the hypothesized associations between periodontitis severity and neurocognitive/neuroimaging outcomes, and the related microbiome/cytokine and morphometrics components (Introduction, end of section).

2 “…clearly present formal hypotheses for their primary objective (e.g., expected direction of the association between periodontal severity and neurocognitive/neuroimaging outcomes), and/or

• explicitly state that all planned analyses are exploratory and clearly distinguish any confirmatory components from exploratory ones, if applicable.”

Thank you for highlighting this important point. We agree that the protocol should clearly distinguish confirmatory elements from exploratory analyses. Accordingly, we revised the Statistical Analysis Plan to (i) state a priori directional hypotheses for the primary objective, specifically, that greater periodontitis severity will be associated with poorer cognitive performance (lower MoCA) and a higher neuroimaging burden of cerebral small vessel disease (higher total SVD score), and that periodontal severity will track with more dysbiotic subgingival microbiome profiles and a more pro-inflammatory gingival crevicular-fluid cytokine signature and (ii) explicitly designate the remaining components (e.g., mediation/moderation and morphometric comparisons) as exploratory and hypothesis-generating.

3. Provide a sample size calculation or justification for the primary objective, including key assumptions.

Thank you for this request. Because this protocol is designed as a census-based survey embedded in the Atahualpa Project, the planned sample size is primarily determined by the size of the source population (all 410 registered residents aged ≥60 years will be invited). We added a “Sample size considerations” paragraph to the Statistical Analysis Plan. For descriptive estimates, using finite population correction (N=410), 95% confidence, 5% absolute precision, and p=0.50, the minimum required sample is 199 participants. For the primary association analyses, we additionally report power-based considerations showing that an analyzable sample in the ~200–410 range provides adequate power to detect small effects on continuous outcomes (e.g., correlations around r≈0.20 with n≈200, and r≈0.17 with n≈410), and we note strategies to ensure model stability (prespecified covariates and dimension reduction for biomarker panels).

4. Provide more detail on examiner calibration and intra-/inter-examiner reliability, particularly for periodontal and dental assessments.

Thank you for this suggestion. We agree that examiner calibration and measurement reliability should be described in greater detail to strengthen reproducibility. Accordingly, we expanded the Methods section to specify pre-field training and calibration procedures for the oral-diagnosis and periodontal examinations, and we added a plan to assess intra- and inter-examiner agreement in a subset of participants using duplicate examinations (including ICCs for continuous periodontal measures such as probing depth and clinical attachment loss, and kappa statistics for categorical measures such as bleeding on probing, furcation involvement, and stage/grade classification), with retraining and repeat calibration if agreement does not meet prespecified targets.

5. The Data Availability statement is not compliant/insufficiently specific; clarify where data will be deposited, access conditions, and any justified restrictions.

Thank you for noting this issue. We revised the Data Availability/dissemination section to comply with PLOS ONE’s data policy. Because this is a study protocol, no data have yet been generated. We now state explicitly that, upon publication of the first results derived from this protocol, the minimal de-identified participant-level dataset underlying the findings will be made available as Supporting Information files, with direct identifiers and any linkage file excluded. If any specific variable is deemed to pose a non-negligible risk of re-identification in this small community, access to an additional de-identified dataset will be provided upon reasonable request, subject to approval by the Human Research Ethics Committee of Kennedy Clinic (CEISH).

6. Minor typographical/grammatical errors; proofread the manuscript.

Thank you for this suggestion. We carefully proofread the manuscript and corrected typographical and minor grammatical issues throughout the text to improve clarity and readability. These edits are reflected in the tracked-changes version of the revised manuscript.

Reviewer 3

1. Explain how the authors will ensure statistical power for the planned analyses.

Thank you for this comment. We have added a “Sample size and power considerations” paragraph to the Population and eligibility. Because this is a census-based survey (all 410 Atahualpa residents aged ≥60 years will be invited), sample size is primarily determined by the source population and participation. We now report (i) a finite-population, 95% confidence/5% precision calculation for descriptive estimates (minimum ~199 participants, p=0.50) and (ii) power-based considerations for the primary association analyses, indicating that this range provides adequate power to detect small effects on continuous outcomes (e.g., MoCA). We also specify strategies to ensure stable multivariable models (prespecified covariates and dimension reduction for biomarker panels).

2. For microbiome analyses, account for potential confounders such as antimicrobial/antifungal use.

Thank you for this important point. We revised the Methods to explicitly capture recent exposures that can affect the oral microbiome, including systemic antibiotics and antifungals as well as topical oral antimicrobials (e.g., chlorhexidine), recording timing and duration. We also updated the Statistical Analysis Plan to prespecify sensitivity analyses excluding participants with recent antimicrobial/antifungal exposure (e.g., within the prior 3 months) and/or adjustment for these exposures as covariates in microbiome-related analyses.

3. Given the cohort context, include quality-of-life assessment and a dietary questionnaire.

Thank you for this suggestion. We clarify that oral health–related quality of life is assessed in this protocol using GOHAI at Station 6. Regarding diet, this study is embedded within the long-standing Atahualpa Project cohort, which has already collected dietary information in prior assessments (including oily-fish intake); these variables will be incorporated as covariates in the planned analyses. Implementing an additional comprehensive dietary questionnaire is outside the scope of this oral-health protocol, but we acknowledge its value and consider it for future extensions if greater dietary granularity is required.

4. Describe how the plaque index will be performed (needs better detail).

Thank you for this comment. We expanded the Methods to describe plaque assessment in detail. Specifically, we will use the O’Leary Plaque Control Record (PCR): after plaque disclosure, we will record the presence/absence of plaque on four surfaces per tooth (mesial, distal, facial/buccal, and lingual/palatal) for all teeth present and express the score as the percentage of examined tooth surfaces with plaque, excluding missing teeth from the denominator. We also prespecified PCR as an oral-hygiene covariate in analyses of periodontal inflammation and related biomarker outcomes, given the established relationship between plaque accumulation and periodontal inflammation/disease.

5. For all clinical measures, provide more depth and operational details.

Thank you for this comment. We expanded the Methods to provide greater detail and operational definitions for all main clinical measurements. Specifically, we now describe full-mouth periodontal charting at six sites per tooth, define PD, REC and CAL (including how CAL is derived), specify BOP recording, and clarify how furcation involvement and tooth mobility are classified. We also explain how CBCT-derived bone loss will be used to support periodontal staging (including the operationalization of percentage bone loss), improving reproducibility and transparency.

6. Justify why the microbial community will be assessed only at the deepest site.

Thank you for this comment. We clarified the rationale for sampling the deepest site per selected tooth. Deeper pockets concentrate the subgingival biofilm most relevant to periodontitis and increase the likelihood of capturing dysbiotic periodontal communities. This strategy also standardizes sampling across participants and is feasible within the station-based clinical workflow. Importantly, we pool samples from four teeth (two incisors and two molars) to obtain a participant-level profile across tooth types/quadrants, and we record clinical parameters (e.g., probing depth and bleeding on probing) at sampled sites to contextualize the microbiome data and support sensitivity analyses.

7. Data management and sharing procedures should be clearer.

Thank you for this comment. We revised the manuscript to clarify both data management and data sharing. In the Methods, we now describe how clinical, and questionnaire data will be captured using standardized electronic databases with quality-control checks, stored securely with role-based access, and de-identified for analysis, with any linkage file stored separately under restricted access. We also revised the Data Availability statement to comply with PLOS ONE policy: because this is a protocol, no data has yet been generated; upon publication of the first results derived from this protocol, the minimal de-identified dataset underlying the findings will be provided as Supporting Information files, with direct identifiers excluded. If any variable is judged to pose a non-negligible risk of re-identification, access to an additional de-identified dataset will be provided upon reasonable request, subject to approval by the Human Research Ethics Committee of Kennedy Clinic (CEISH).

---

## [Editor Report · Decision Letter 1]

27 Jan 2026

Study protocol: Dental and periodontal characteristics of older adults in Atahualpa, Ecuador within the Atahualpa Project cohort.

PONE-D-25-59284R1

Dear Dr. Armijos Briones,

We’re pleased to inform you that your manuscript has been judged scientifically suitable for publication and will be formally accepted for publication once it meets all outstanding technical requirements.

Kind regards,

Vinayak M. Joshi, MS, PhD

Academic Editor

PLOS One